# When should agents explore?

## Abstract

Exploration remains a central challenge for reinforcement learning (RL). Virtually all existing methods share the feature of a *monolithic* behaviour policy that changes only gradually (at best). In contrast, the exploratory behaviours of animals and humans exhibit a rich diversity, namely including forms of *switching* between modes. This paper presents an initial study of mode-switching, non-monolithic exploration for RL. We investigate different modes to switch between, at what timescales it makes sense to switch, and what signals make for good switching triggers. We also propose practical algorithmic components that make the switching mechanism adaptive and robust, which enables flexibility without an accompanying hyper-parameter-tuning burden. Finally, we report a promising and detailed analysis on Atari, using two-mode exploration and switching at sub-episodic time-scales.

## 1 Introduction

The trade-off between exploration and exploitation is described as the crux of learning and behaviour across many domains, not just reinforcement learning [Sutton and Barto, 2018], but also in decision making [Cohen et al., 2007], evolutionary biology [Cremer et al., 2019], ecology [Kembro et al., 2019], neuroscience (e.g., focused versus diffuse search in visual attention [Wolfe et al., 1989], dopamine regulations [Chakroun et al., 2020]), cognitive sciences [Hills et al., 2015], as well as psychology and psychiatry [Addicott et al., 2017]. In a nutshell, exploration is about the balance between taking the familiar choice that is known to be rewarding and learning about unfamiliar options of uncertain reward, but which could ultimately be more valuable than the familiar options.

Ample literature has studied the question of *how much* to explore, that is how to set the overall trade-off (and how to adjust it over the course of learning) [Jaksch et al., 2010, Cappé et al., 2013, Lattimore and Szepesvári, 2020, Thrun, 1992], and the question of *how* to explore, namely how to choose exploratory actions (e.g., randomly, optimistically, intrinsically motivated, or otherwise) [Schmidhuber, 1991, Oudeyer and Kaplan, 2009, Linke et al., 2019]. In contrast, the question of *when* to explore has been studied very little, possibly because it does not arise in bandit problems, where a lot of exploration methods are rooted. The 'when' question and its multiple facets are the subjects of this paper. We believe that addressing it could lead to more *intentional* forms of exploration.

Consider an agent that has access to two *modes* of behaviour, an 'explore' mode and an 'exploit' mode (e.g., a random policy and a greedy policy, as in $\varepsilon$-greedy). Even when assuming that the overall proportion of exploratory steps is fixed, the agent still has multiple degrees of freedom: it can explore more at the beginning of training and less in later phases; it may take single exploratory steps or execute prolonged periods of exploration; it may prefer exploratory steps early or late within an episode; and it could trigger the onset (or end) of an exploratory period based on various criteria. Animals and humans exhibit non-trivial behaviour in all of these dimensions, presumably encoding useful inductive biases that way [Power, 1999]. Humans make use of multiple effective strategies, such as selectively exploring options with high uncertainty (a form of directed, or information-seeking exploration), and increasing the randomness of their choices when they are more uncertain [Gershman,

2018, Gershman and Tzovaras, 2018, Ebitz et al., 2019]. Monkeys use directed exploration to manage explore-exploit trade-offs, and these signals are coded in motivational brain regions [Costa et al., 2019]. Patients with schizophrenia register changes in directed exploration and experience low-grade inflammation when shifting from exploitation to random exploration [Waltz et al., 2020, Cathomas et al., 2021]. This diversity is what motivates us to study which of these can benefit RL agents in turn, by expanding the class of exploratory behaviours beyond the commonly used *monolithic* ones (where modes are merged homogeneously in time).

## 2 Methods

The objective of an RL agent is to learn a policy that maximises external reward. At the high level, it achieves this by interleaving two processes: generating new experience by interacting with the environment using a behaviour policy (exploration) and updating its policy using this experience (learning). As RL is applied to increasingly ambitious tasks, the challenge for exploration becomes to keep producing *diverse* experience, because if something has not been encountered, it cannot be learned. Our central argument is therefore simple: a monolithic, time-homogeneous behaviour policy is strictly less diverse than a heterogeneous mode-switching one, and the former may hamstring the agent's performance. As an illustrative example, consider a human learning how to ride a bike (explore), while maintaining their usual happiness through food, sleep, work (exploit): there is a stark contrast between a monolithic, time-homogeneous behaviour that interleaves a twist of the handlebar or a turn of a pedal once every few minutes or so, and the mode-switching behaviour that dedicates prolonged periods of time exclusively to acquiring the new skill of cycling.

### 2.1 Exploration modes

While the choice of behaviour in pure exploit mode is straightforward, namely the greedy pursuit of external reward (or best guess thereof), denoted by $\mathcal{G}$, there are numerous viable choices for behaviour in a pure explore mode (denoted by $\mathcal{X}$). In this paper we consider two standard ones: $\mathcal{X}_U$, the naive uniform random policy, and $\mathcal{X}_I$, an intrinsically motivated behaviour that exclusively pursues a novelty measure based on random network distillation (RND, [Burda et al., 2018]). See Section 4 and Appendix B for additional possibilities of $\mathcal{X}$. In this paper we choose fixed behaviours for these modes, and focus solely on the question of *when* to switch between them. In our setting, overall proportion of exploratory steps (the *how much*), denoted by $p_\mathcal{X}$, is not directly controlled but derives from the *when*.

### 2.2 Granularity

An exploration *period* is an uninterrupted sequence of steps in explore mode. We consider four choices of temporal granularity for exploratory periods, also illustrated on Figure 1:

**Step-level** exploration is the most common scenario, where the decision to explore is taken independently at each step, affecting one action.[1] The canonical example is $\varepsilon$-greedy (Fig.1:C).

**Experiment-level** exploration is the other extreme, where all behaviour during training is produced in explore mode, and learning is off-policy (the greedy policy is only used for evaluation). This scenario is also very common, with most forms of intrinsic motivation falling into this category, namely pursuing reward with an intrinsic bonus throughout training (Fig.1:A).[2]

**Episode-level** exploration is the case where the mode is fixed for an entire episode at a time (e.g., training games versus tournament matches in a sport), see Fig.1:B. This has been investigated for simple cases, where the policy's level of stochasticity is sampled at the beginning of each episode [Horgan et al., 2018, Kapturowski et al., 2019, Zha et al., 2021].

**Intra-episodic** exploration is what falls in-between step- and episode-level exploration, where exploration periods last for multiple steps, but less than a full episode. This is the least commonly studied scenario, and will form the bulk of our investigations (Fig.1:D,E,F,G).

---

[1]The length of an exploratory period tends to be short, but it can be greater than 1, as multiple consecutive step-wise decisions to explore can create longer periods.

[2]Note that it is also possible to interpret $\varepsilon$-greedy as experiment-level exploration, where the $\mathcal{X}$ policy is fixed to a noisy version of $\mathcal{G}$.

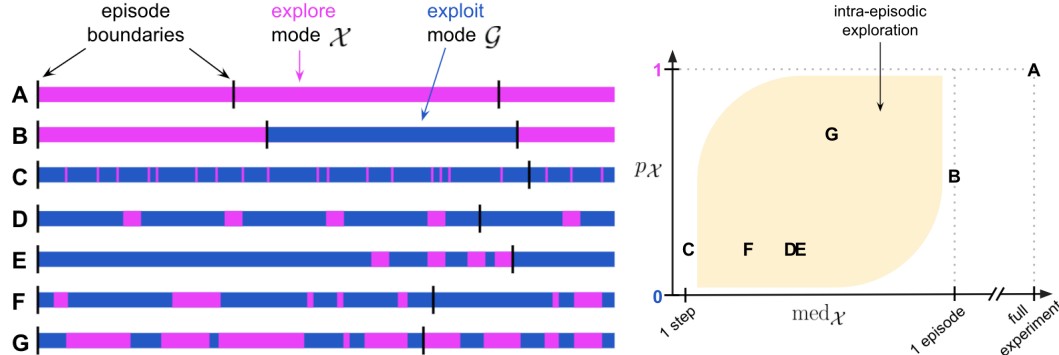

Figure 1: Illustration of different types of temporal structure for two-mode exploration. **Left**: Each line A-G depicts an excerpt of an experiment (black lines show episode boundaries, experiment continues on the right), with colour denoting the active mode (blue is exploit, magenta is explore). **A** is of experiment-level granularity, **B** episode-level, **C** step-level, and **D-G** are of intra-episodic exploration granularity. **Right**: The same examples, mapped onto a characteristic plot of summary statistics: overall exploratory proportion $p_\mathcal{X}$ versus typical length of an exploratory period $\mathrm{med}_\mathcal{X}$. The yellow-shaded area highlights the intra-episodic part of space studied in this paper (some points are not realisable, e.g., when $p_\mathcal{X} \approx 1$ then $\mathrm{med}_\mathcal{X}$ must be large). **C, D, E, F** share the same $p_\mathcal{X} \approx 0.2$, while interleaving exploration modes in different ways. **D** and **E** share the same $\mathrm{med}_\mathcal{X}$ value, and differ only on whether exploration periods are spread out, or happen toward the end of episode.

We denote the length of an exploratory period by $n_\mathcal{X}$ (and similarly $n_\mathcal{G}$ for exploit mode). To characterise granularity, our summary statistic of choice is $\mathrm{med}_\mathcal{X} := \mathrm{median}(n_\mathcal{X})$. Note that there are two possible units for these statistics: the raw steps or the proportion of the episode length $L$. The latter has different (relative) semantics, but may be more appropriate when episode lengths vary widely across training. We denote it as $\mathrm{rmed}_\mathcal{X} := \mathrm{median}(n_\mathcal{X}/L)$.

## 2.3 Switching for intra-episodic exploration

Granularity is but the coarsest facet of the 'when' question, but more precise intra-episode timings (when to start and when to stop an exploratory period) are important aspects too.

**Blind switching**    The simplest type of switching mechanism does not take state or time into account (thus we call it *blind*), and is only concerned with producing switches at some desired time resolution. It can be implemented deterministically through a counter (e.g., enter explore mode after 100 exploit mode steps), or probabilistically (e.g., at each step, enter explore mode with probability 0.01). Its expected duration can be parameterised in terms of raw steps, or in terms of fractional episode length. The opposite of blind switching is *informed* switching, as discussed in Section 2.4.

**Asymmetry**    In general, the mechanism for entering the explore mode can differ from the one for exiting it (to enter the exploit mode), and this is crucial to obtain flexible overall amounts of exploration – if switching were symmetric, the proportion would be $p_\mathcal{X} \approx 0.5$.

**Starting mode**    When periods last for a significant fraction of episode length, it also matters how the sequence is initialised, i.e., whether an episode starts in explore or in exploit mode, or more generally, whether the agent explores more early in an episode or more later on. It is conceivable that the best choice among these is domain dependent (see Figure 6): in most scenarios, the states at the beginning of an episode have been visited many times, thus starting with exploit mode can be beneficial; in other domains however, early actions may disproportionately determine the available future paths (e.g., build orders in StarCraft [Churchill and Buro, 2011]).

## 2.4 Informed switching with triggers

Going beyond blind switching opens up another rich set of design choices. We decompose the mechanism into two parts. First, a scalar *trigger* signal is produced by the agent at each step, based on its current information – drawing inspiration from human behaviour, the triggering signal is intended

113 to be a proxy for uncertainty [Schulz et al., 2019]. Second, a binary switching decision is taken based
114 on the trigger signal, for example by comparing it to a threshold. Again, the type of trigger and its
115 configuration will in general not be symmetric between entering and exiting an exploratory period.

**Value promise trigger**  To keep this paper focused, we will look at one such trigger, dubbed 'value
117 promise discrepancy' (see Appendix B for additional competitive variants). This is an online proxy
118 of how much of the reward that the agent's past value estimate promised ($k$ steps ago) have actually
119 come about. The intuition is that in uncertain parts of state space, this discrepancy will generally be
120 larger than when everything goes as expected. Formally,

$$D_{\mathrm{promise}}(t - k, t) := \left| V(s_{t-k}) - \sum_{i=0}^{k-1} \gamma^i R_{t-i} - V(s_t) \right|$$

121 where $V(s)$ is the agent's value estimate at state $s$, $R$ is the reward, and $\gamma$ is a discount factor.

**Homeostasis**  In practice, the scales of trigger signals may vary substantially across domains, and
123 across training time, for example, the magnitude of $D_{\mathrm{promise}}$ will depend on reward scales and
124 density, and can decrease over time as accuracy improves (the signals could also be noisy). This
125 means that naively setting a threshold hyper-parameter is impractical. For a simple remedy, we have
126 taken inspiration from neuroscience [Turrigiano and Nelson, 2004] to add homeostasis to the binary
127 switching mechanism, which tracks recent values of the signal and adapts the threshold for switching
128 so that a specific average *target rate* is obtained. This functions as an adaptive threshold, making
129 tuning straightforward because the target rate of switching can be configured independently of the
130 scales of the trigger signal. See Appendix A for the details of the implementation.

### 2.5   Adaptation instead of tuning

132 Our approach introduces additional flexibility to the exploration process, even when holding the
133 specifics of the learning algorithm and the exploration mode fixed. The two main added dimensions
134 are when (or how often) to enter explore mode, and when (or how quickly) to exit it. To avoid this
135 becoming a hyper-parameter tuning burden, we propose to follow [Schaul et al., 2019] and [Badia
136 et al., 2020a], and delegate the adaptation of these settings to a meta-controller (implemented as a
137 non-stationary multi-armed bandit that maximises episodic return). As an added benefit, the 'when'
138 of exploration can now become adaptive to both the task, and the stage of learning.

## 3   Results

140 The design space we propose contains a number of atypical ideas for how to structure exploration.
141 For this reason, we opted to keep the rest of our experimental setup very conventional, and include
142 multiple comparable baselines, ablations and variations.

**Setup: R2D2 on Atari**  We conduct our investigations on a subset of games of the Atari Learning
144 Environment [Bellemare et al., 2013], a common benchmark for the study of exploration.  All
145 experiments are conducted across 7 games (FROSTBITE, GRAVITAR, H.E.R.O., MONTEZUMA'S
146 REVENGE, MS. PAC-MAN, PHOENIX, STAR GUNNER), the first 5 of which are classified as hard
147 exploration games [Bellemare et al., 2016], using 3 seeds per game. For our agent, we use the R2D2
148 architecture [Kapturowski et al., 2019], which is a modern, distributed version of DQN [Mnih et al.,
149 2015] that employs a recurrent network to approximate its Q-value function.  This is a common
150 basis used in exploration studies, e.g., [Dabney et al., 2020, Badia et al., 2020b,a]. The only major
151 modification to conventional R2D2 is its exploration mechanism, where instead we implement all the
152 variants of mode-switching introduced in Section 2. Separately from the experience collected for
153 learning, we run an evaluator process that assesses the performance of the current greedy policy. This
154 is what we report in all our performance curves (see Appendix A for more details).

**Baselines**  There are a few simple baselines worth comparing to, namely the pure explore mode
156 ($p_\mathcal{X} = 1$, Fig.1:A) and the pure exploit mode ($p_\mathcal{X} = 0$), as well as the step-wise interleaved $\varepsilon$-greedy
157 execution (Fig.1:C), where $p_\mathcal{X} = 0.01 = \varepsilon$ (without additional episodic or intra-episodic structure).
158 Given its wide adoption in well-tuned prior work, we expect the latter to perform well overall.

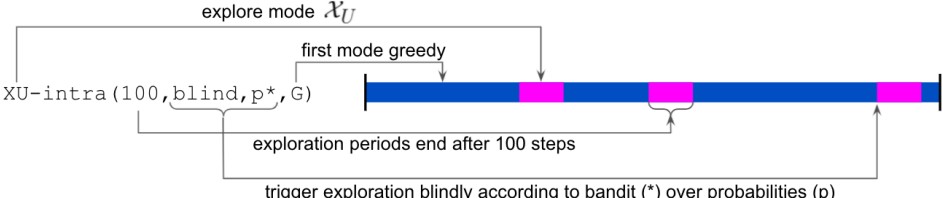

Figure 2: Illustrating the space of design decisions for intra-episodic exploration.

The fourth baseline picks a mode for an entire episode at a time (Fig.1:B), with the probability of picking $\mathcal{X}$ being adapted by a bandit meta-controller. We denote these as `experiment-level-X`, `experiment-level-G`, `step-level-0.01` and `episode-level-*` respectively. For each of these, we have a version with uniform ($\mathcal{X}_U$) and intrinsic ($\mathcal{X}_I$) explore mode.

## 3.1 Variants of intra-episodic exploration

As discussed in Section 2, there are multiple dimensions along which two-mode intra-episodic exploration can vary. The concrete ones for our experiments are:

- Explore mode: uniform random $\mathcal{X}_U$, or RND intrinsic reward $\mathcal{X}_I$ (denoted `XU` and `XI`).

- Explore duration ($n_\mathcal{X}$): this can be a fixed number of steps $(1, 10, 100)$, or one of these is adaptively picked by a bandit (denoted by `*`), or the switching is symmetric between entering end exiting explore mode (denoted by `=`).

- Trigger type: either `blind` or `informed` (based on value promise, see Section 2.4).

- Exploit duration ($n_\mathcal{G}$): for blind triggers, the exploit duration can be parameterised by fixed number of steps $(10, 100, 1000, 10000)$, indirectly defined by a probability of terminating $(0.1, 0.01, 0.001, 0.0001)$, or adaptively picked by a bandit over these choices (denoted by `n*` or `p*`, respectively). For informed triggers, the exploit duration is indirectly parameterised by a target rate in $(0.1, 0.01, 0.001, 0.0001)$, or a bandit over them (`p*`), which is in turn transformed into an adaptive switching threshold by homeostasis (Section 2.4).

- Starting mode: $\mathcal{G}$ greedy (default) or $\mathcal{X}$ explore (denoted by `G` or `X`).

We can concisely refer to a particular instance by a tuple that lists these choices. For example, `XU-intra(100,informed,p*,X)` denotes uniform random exploration $\mathcal{X}_U$, with fixed 100-step explore periods, triggered by the value-promise signal at a bandit-determined rate, and starting in explore mode. See Figure 2 for an illustration.

## 3.2 Performance results

We start by reporting overall performance results, to reassure the reader that our method is viable (and convince them to keep reading the more detailed and qualitative results in the following sections). Figure 3 shows performance across 7 Atari games according to two human-normalised aggregation metrics (mean and median), comparing one form of intra-episodic exploration to all the baselines, separately for each explore mode ($\mathcal{X}_U$ and $\mathcal{X}_I$). The headline result is that intra-episodic exploration improves over both step-level and episode-level baselines (as well as the pure experiment-level modes that we would not expect to be very competitive). The full learning curves per game are found in the appendix, and show scores on hard exploration games like MONTEZUMA'S REVENGE or PHOENIX that are also competitive in absolute terms (at our compute budget of 1B frames).

Note that there is a subtle difference to the learning setups between $\mathcal{X}_U$ and $\mathcal{X}_I$, as the latter requires training a separate head to estimate intrinsic reward values. This is present even in pure exploit mode, where it acts as an auxiliary task only [Jaderberg et al., 2016], hence the differences in pure greedy curves in Figure 3. For details, see Appendix A.

## 3.3 Diversity results

In a study like ours, the emphasis is not on measuring raw performance, but rather on characterising the diversity of behaviours arising from the spectrum of proposed variants. A starting point is to

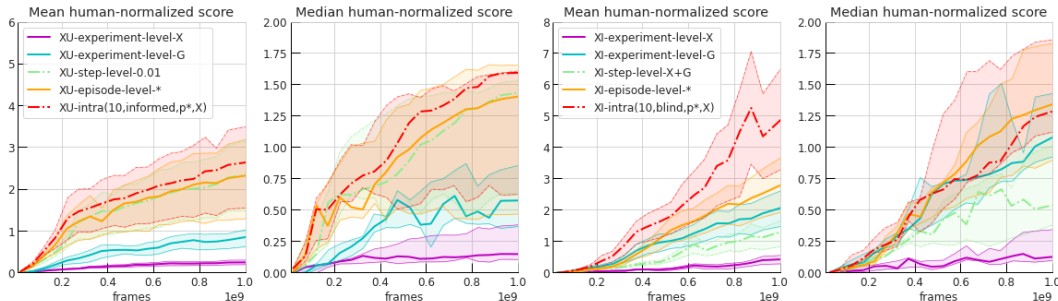

Figure 3: Human-normalized performance results aggregated over 7 Atari games and 3 seeds, comparing the four levels of exploration granularity. **Left two**: uniform explore mode $\mathcal{X}_U$. **Right two**: RND intrinsic reward explore mode $\mathcal{X}_I$. In each case, the baselines are pure modes $\mathcal{X}$ and $\mathcal{G}$, step-level switching with $\varepsilon$-greedy, and episodic switching (with a bandit-adapted proportion). In each setting, intra-episodic exploration is on par or better than the baselines.

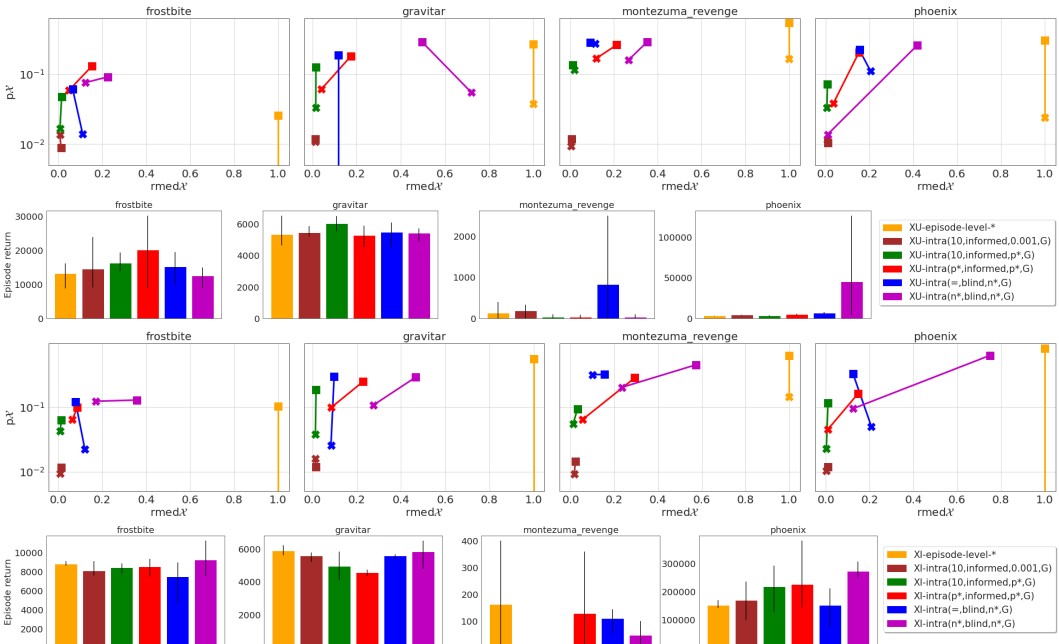

Figure 4: **Rows 1 and 3**: Summary characteristics $p_{\mathcal{X}}$ and $\mathrm{rmed}_{\mathcal{X}}$ of induced exploration behaviour, for different variants of intra-episodic exploration (and an episodic baseline for comparison), on a subset of 4 Atari games. Bandit adaptation can change these statistics over time, hence square and cross markers show averages over first and last $10\%$ of training, respectively. **Rows 2 and 4**: Corresponding final scores (averaged over final $10\%$ of training). Error bars show the span between min and max performance across 3 seeds. Note how different variants cover different parts of characteristic space, and how the bandit adaptation shifts the statistics into different directions for different games. See main text for further discussion of these results and Appendix C for other games and variants.

return to Figure 1 (right), and assess how much of the previously untouched space is now filled by intra-episodic variants, and how the 'when' characteristics translate into performance. Figure 4 answers these questions, and raises some new ones. First off, the raw amount of exploration $p_{\mathcal{X}}$ is not a sufficient predictor of performance, implying that the temporal structure matters. It also shows substantial bandit adaptation at work: compare the exploration statistics at the start (squares) and end-points of training (crosses), and how these trajectories differ per game; a common pattern is that reducing $p_{\mathcal{X}}$ far below $0.5$ is needed for high performance. Interestingly, these adaptations are similar between $\mathcal{X}_U$ and $\mathcal{X}_I$, despite very different explore modes (and differing performance results). We would expect prolonged intrinsic exploration periods to be more useful than prolonged random ones, and indeed, comparing the high-$\mathrm{rmed}_{\mathcal{X}}$ variant (purple) across $\mathcal{X}_U$ and $\mathcal{X}_I$, it appears more

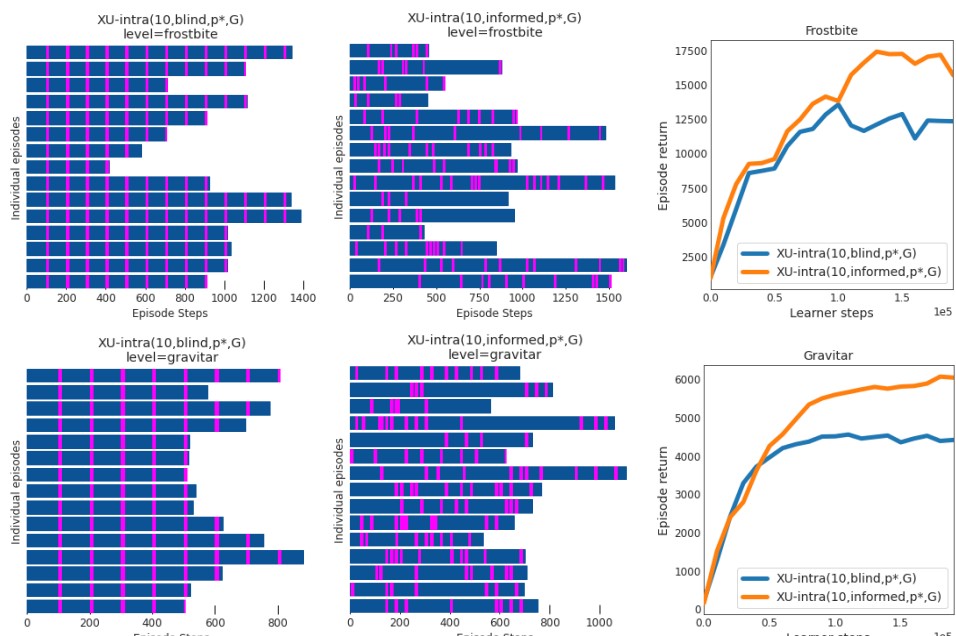

Figure 5: **Left and center**: Illustration of detailed temporal structure within individual episodes, on FROSTBITE (top) and GRAVITAR (bottom), contrasting two trigger mechanisms. Each subplot shows 15 randomly selected episodes (one per row) that share the same overall exploration amount $p_{\mathcal{X}} = 0.1$. Each vertical bar (magenta) represents an exploration period of fixed length $n_{\mathcal{X}} = 10$; each blue chunk represents an exploitation period. **Left**: blind, step-based trigger leads to equally spaced exploration periods. **Center**: a trigger signal informed by value promise leads to very different within-episode patterns, with some parts being densely explored, and others remaining in exploit mode for very long. **Right**: the corresponding learning curves show a clear performance benefit for the informed trigger variant (orange) in this particular setting. Appendix C has similar plots for many more variants and games.

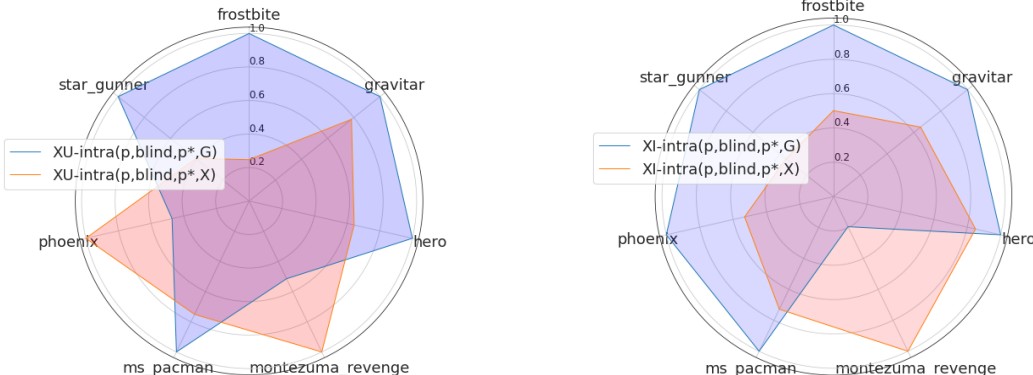

Figure 6: Starting mode effect. Final mean episode return for two blind intra-episode experiments that differ only in start mode, greedy (blue) or explore (orange). Scores are normalised so that 1 is the maximum result across the two start modes. Either choice can reliably boost or harm performance, depending on the game. **Left**: uniform explore mode $\mathcal{X}_U$. **Right**: intrinsic reward explore mode $\mathcal{X}_I$.

beneficial for the latter. Zooming in on specific games, a few results stand out: in $\mathcal{X}_U$ mode, the only variant that escapes the inherent local optimum of PHOENIX is the blind, doubly adaptive one (purple), with the bandits radically shifting the exploration statistics over the course of training. In contrast, the best results on MONTEZUMA'S REVENGE are produced by the symmetric trigger variant (blue), which is forced to retain a high $p_{\mathcal{X}}$. Finally, FROSTBITE is the one game where an informed trigger (red) clearly outperforms its blind equivalent (purple).

These insights are still limited to summary statistics, so Figure 5 looks in more depth at the detailed temporal structure within episodes (as in Figure 1, left). Here the main comparison is between blind

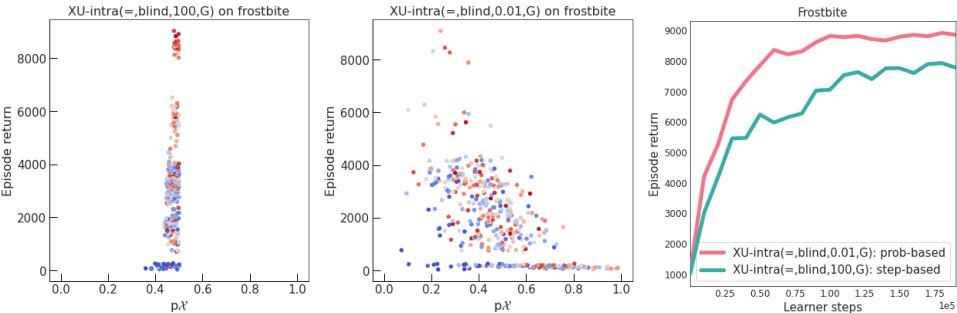

Figure 7: **Left and center**: Contrasting the behavioural characteristics between two forms of blind switching, step-based (left) and probabilistic (center), on the example of FROSTBITE. Each point is an actor episode, with colour indicating time in training (blue for early, red for late). Note the higher diversity of $p_\mathcal{X}$ when switching probabilistically. **Right**: Corresponding performance curves indicate that the probabilistic switching (red) has a performance benefit, possibly because it creates the opportunity for 'lucky' episodes with much less randomness in a game where random actions can easily kill the agent. For more games, please see the Appendix C.

and informed triggers, illustrating that the characteristics of the fine-grained within-episode structure can differ massively, despite attaining the same high-level statistics $p_\mathcal{X}$ and $\mathrm{med}_\mathcal{X}$. We can see quite a lot of variation in the trigger structure – the moments we enter exploration are not evenly spaced anymore. As a bonus, the less rigid structure of the informed trigger (and possibly the more carefully chosen switch points) end up producing better performance too.

Figure 6 sheds light on a complementary dimension, differentiating the effects of starting in explore or exploit mode. In brief, each of these can be consistently beneficial in some games, and consistently harmful in others. Another observation here is the dynamics of the bandit adaptation: when starting in exploit mode, it exhibits a preference for long initial exploit periods in many games (up to 10000 steps), but that effect vanishes when starting in explore mode (see also Appendix C). More subtle effects arise from the choice of parameterisation of switching rates. Figure 7 shows a stark qualitative difference on how probabilistic switching differs from step-count based switching, with the former spanning a much wider diversity of outcomes, which improves performance.

### 3.4 Take-aways

Summarising the empirical results in this section, two messages stand out. First, there seems to be a sweet spot in terms of temporal granularity, and intra-episodic exploration is the right step towards finding it. Second, the vastly increased design space of our proposed family of methods gives rise to a large diversity of behavioural characteristics; and this diversity is not superficial, it also translates to meaningful performance differences, with different effects in different games, which cannot be reduced to simplistic metrics, such as $p_\mathcal{X}$. In addition, we provide some sensible rules-of-thumb for practitioners willing to join us on the journey of intra-episodic exploration. In general, it is useful to let a bandit figure out the precise settings, but it is worth curating its choices to at most a handful. Jointly using two bandits across factored dimensions is very adaptive, but can sometimes be harmful when they decrease the signal-to-noise ratio in each other's learning signal. Finally, the choice of the uncertainty-based trigger should be informed by the switching modes (see Appendix B for details).

## 4 Discussion

**Time-based exploration control** The emphasis of our paper has been on the potential benefits of heterogeneous temporal structure in mode-switching exploration. But there is another, more mundane potential advantage over monolithic approaches: it may be easier and more natural to tune hyper-parameters related to an explicit exploration budget (e.g., via $p_\mathcal{X}$) than to tune an intrinsic reward coefficient, especially if extrinsic reward scales change across tasks or across time, and if the non-stationarity of the intrinsic reward affects its overall scale.

**Diversity for diversity's sake** One role of a general-purpose exploration method is to allow an agent to get off the ground in a wide variety of domains. While this may clash with sample-efficient

learning on specific domains, we believe that the former objective will come to dominate in the long run. In this light, methods that exhibit more diverse behaviour are preferable for that reason alone, because they are more likely to escape local optima or misaligned priors.

**Related work**   While not the most common approach to exploration in RL, we are aware of some notable work that has investigated non-trivial temporal structure. The $\epsilon z$-greedy algorithm [Dabney et al., 2020] is inspired by Levy flights in nature [Baronchelli and Radicchi, 2013] and initiates contiguous chunks of directed behaviour ('flights') with the length sampled from a heavy-tailed distribution. In contrast to our proposal, these flights act with a single constant action, instead of invoking an explore mode. [Campos et al., 2021] pursue a similar idea, but with flights along pre-trained coverage policies, while [Ecoffet et al., 2021] chain a 'return-to-state' policy to an explore mode. Maybe closest to our $\mathcal{X}_I$ setting is [Bagot et al., 2020], where periods of intrinsic reward pursuit are explicitly invoked by the agent. Exploration with *gradual* change instead of abrupt mode switches, appears generally at long time-scales, such as when pursuing intrinsic rewards [Schmidhuber, 2010, Oudeyer and Kaplan, 2009], but can also be effective at shorter time-scales e.g., Never-Give-Up [Badia et al., 2020b]. Related work on the question of which states to prefer for exploratory decisions [Tokic, 2010] tends to not consider starting prolonged exploratory periods.

**Relation to options**   Ideas related to switching behaviours at intra-episodic time-scales are well-known outside of the context of exploration, the best-known framework being *options* in hierarchical RL, where the goal is to chain together a sequence of sub-behaviours into a reward-maximising policy [Sutton et al., 1999, Mankowitz et al., 2016]; but some work has looked at using options for exploration too [Jinnai et al., 2019a, Bougie and Ichise, 2021]. In its full generality, the options framework is a substantially more ambitious endeavour than our proposal, as it requires learning a full state-dependent hierarchical policy that picks which option to start (and when), as well as jointly learning the options themselves.

**Limitations**   Our proposed approach inherits many of the challenges that are typical for exploration methods, such as sample efficiency or trading off risk. An aspect that is particular to the intra-episode switching case is the different nature of the off-policy-ness. The resulting effective policy can produce state distributions that differ substantially from those of either of the two base mode behaviours that are being interleaved. It can potentially visit parts of the state space that neither base policy would reach if followed from the beginning of the episode. While a boon for exploration, this might pose a challenge to learning, as it could require off-policy corrections that treat those states differently and do not only correct for differences in action space. We leave this as an intriguing consideration for future work; this paper does not use any non-trivial off-policy correction (see Appendix A).

**Future work**   With the dimensions laid out in Section 2, it should be clear that this paper can but scratch the surface. We see numerous opportunities for future work, on some of which we already carried out initial investigations, see Appendix B. For starters, there is no inherent need to restrict the mechanism to just two modes: A richer form of exploration could switch between exploit, explore, novelty and mastery [Thomaz and Breazeal, 2008], or between many diverse forms of exploration (such as different levels of optimism [Derman et al., 2020, Moskovitz et al., 2021]). It is also conceivable to switch less abruptly; for example, if both exploit- and explore-mode behaviours are induced by a reward function, a Q-value-based agent with successor features [Barreto et al., 2017, Borsa et al., 2019] could interpolate between them to make switching more gradual [Barreto et al., 2019]. Triggers are another aspect that could be expanded or refined: there are different candidates for estimating uncertainty, such as ensemble discrepancy [Wiering and Van Hasselt, 2008, Buckman et al., 2018], amortised value errors [Flennerhag et al., 2020], or density models [Bellemare et al., 2016, Ostrovski et al., 2017]; also, triggers could be based on other signals that are not derived from uncertainty, such as salience [Downar et al., 2002], minimal coverage [Jinnai et al., 2019a,b], or empowerment [Klyubin et al., 2005, Gregor et al., 2016, Houthooft et al., 2016].

**Conclusion**   We have presented an initial study of intra-episodic exploration, centred on the scenario of switching between an explore and an exploit mode. We hope this has broadened the available forms of temporal structure in behaviour, leading to more diverse, adaptive and intentional forms of exploration, in turn enabling RL to scale to ever more complex domains.

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
