## A  Detailed experimental setup

### A.1  Atari environment

We use a selection of games from the widely used Atari Learning Environment (ALE, [Bellemare et al., 2013]). It is configured to not expose the 'life-loss' signal, and use the full action set (18 discrete actions) for all games (not the per-game reduced effective action spaces). We also use the *sticky*-action randomisation as in [Machado et al., 2018]. Episodes time-out after 108k frames (i.e. 30 minutes of real-time game play).

Differently from most past Atari RL agents following DQN [Mnih et al., 2015], our agent uses the raw $210 \times 160$ RGB frames as input to its value function (one at a time, without frame stacking), though it still applies a max-pool operation over the most recent 2 frames to mitigate flickering inherent to the Atari simulator. As in most past work, an action-repeat of 4 is applied, over which rewards are summed.

### A.2  Agent

The agent used in our Atari experiments is a distributed implementation of a value- and replay-based RL algorithm derived from the Recurrent Replay Distributed DQN (R2D2) architecture [Kapturowski et al., 2019]. This system comprises of a fleet of 120 CPU-based actors (combined with a single TPU for batch inference) concurrently generating experience and feeding it to a distributed experience replay buffer, and a single TPU-based learner randomly sampling batches of experience sequences from replay and performing updates of the recurrent value function by gradient descent on a suitable RL loss.

The value function is represented by a convolutional torso feeding into a linear layer, followed by a recurrent LSTM [Hochreiter and Schmidhuber, 1997] core, whose output is processed by a further linear layer before finally being output via a Dueling value head [Wang et al., 2016]. The exact parameterisation follows the slightly modified R2D2 presented in [Dabney et al., 2020] and [Schaul et al., 2021], see Table 1 for a full list of hyper-parameters. It is trained via stochastic gradient descent on a multi-step TD loss (more precisely, a 5-step Q-learning loss) with the use of a periodically updated target network [Mnih et al., 2015] for bootstrap target computation, using minibatches of sampled replay sequences. Replay sampling is performed using prioritized experience replay [Schaul et al., 2016] with priorities computed from sequences' TD errors following the scheme introduced in [Kapturowski et al., 2019]. As in R2D2, sequences of 80 observations are used for replay, with a prefix of 20 observations used for burn-in. In a slight deviation from the original, our agent uses a fixed replay ratio of 1, i.e. the learner or actors get throttled dynamically if the average number of times a sample gets replayed exceeds or falls below this value; this makes experiments more reproducible and stable.

Actors periodically pull the most recent network parameters from the learner to be used in their exploratory policy. In addition to feeding the replay buffer, all actors periodically report their reward, discount and return histories to the learner, which then calculates running estimates of reward, discount and return statistics to perform return-based scaling [Schaul et al., 2021]. If applicable, the episodic returns from the actors are also sent to the non-stationary bandit(s) that adapt the distribution over exploration parameters (e.g., target ratios $\rho$ or period lengths $n_\chi$). In return, the bandit(s) provide samples from that distribution to each actor at the start of a new episode, as in [Schaul et al., 2019].

Our agent is implemented with JAX [Bradbury et al., 2018], uses the Haiku [Hennigan et al., 2020], Optax [Budden et al., 2020b], Chex [Budden et al., 2020a], and RLax [Hessel et al., 2020] libraries for neural networks, optimisation, testing, and RL losses, respectively, and Reverb [Cassirer et al., 2020] for distributed experience replay.

### A.3  Training and evaluation protocols

All our experiments ran for 200k learner updates. With a replay ratio of 1, sequence length of 80 (adjacent sequences overlapping by 40 observations), a batch size of 64, and an action-repeat of 4 this corresponds to a training budget of $200000 \times 64 \times 40 \times 1 \times 4 \approx 2\text{B}$ environment frames (which

| Neural Network | |
|---|---|
| Convolutional torso channels | $32, 64, 128, 128$ |
| Convolutional torso kernel sizes | $7, 5, 5, 3$ |
| Convolutional torso strides | $4, 2, 2, 1$ |
| Pre-LSTM linear layer units | $512$ |
| LSTM hidden units | $512$ |
| Post-LSTM linear layer units | $256$ |
| Dueling value head units | $2 \times 256$ (separate linear layer for each of value and advantage) |
| **Acting** | |
| Initial random No-Ops | None |
| Sticky actions | Yes (prob 0.25) |
| Action repeats | 4 |
| Number of actors | 120 |
| Actor parameter update interval | 400 environment steps |
| **Replay** | |
| Replay sequence length | 80 (+ prefix of 20 of burn-in) |
| Replay buffer size | $4 \times 10^6$ observations ($10^5$ part-overlapping sequences) |
| Priority exponent | 0.9 |
| Importance sampling exponent | 0.6 |
| Fixed replay ratio | 1 update per sample (on average) |
| **Learning** | |
| Multi-step Q-learning | $k = 5$ |
| Off-policy corrections | None |
| Discount $\gamma$ | 0.997 |
| Reward clipping | None |
| Return-based scaling | as in [Schaul et al., 2021] |
| Mini-batch size | 64 |
| Optimizer & settings | Adam [Kingma and Ba, 2014], learning rate $\eta = 2 \times 10^{-4}$, $\epsilon = 10^{-8}$, momentum $\beta_1 = 0.9$, second moment $\beta_2 = 0.999$ |
| Gradient norm clipping | 40 |
| Target network update interval | 400 updates |
| **RND settings** | |
| Convolutional torso channels | $32, 64, 64$ |
| Convolutional torso kernel sizes | $8, 4, 3$ |
| Convolutional torso strides | $4, 2, 1$ |
| MLP hidden units | 128 |
| Image downsampling stride | $2 \times 2$ |

Table 1: Hyper-parameters and settings.

is less than 10% of the original R2D2 budget). In wall-clock-time, one such experiment takes about 12 hours (while 2 TPUs and 120 CPUs).

For evaluation, a separate actor (not feeding the replay buffer) is running alongside the agent using a greedy policy ($\varepsilon = 0$), and pulling the most recent parameters at the beginning of each episode. We follow standard evaluation methodology for Atari, reporting mean and median 'human-normalised' scores as introduced in [Mnih et al., 2015] (i.e. the episode returns are normalised so that 0 corresponds to the score of a uniformly random policy while 1 corresponds to human performance), as well as the mean 'human-capped' score which caps the per-game performance at human level. Error bars or shaded curves correspond to the minimum and maximum values across these seeds.

## A.4 Random network distillation

The agent setup for the $\mathcal{X}_I$ experiments differs in a few ways from the default described above. First, a separate network is trained via Random Network Distillation (RND, [Burda et al., 2018]), which consists of a simple convnet with an MLP (no recurrence); for detailed settings, see RND section in Table 1. The RND prediction network is updated jointly with the Q-value network, on the same data. The intrinsic reward derived from the RND loss is pursued at the same discount $\gamma = 0.997$ as

the external reward in $\mathcal{G}$. The Q-value network is augmented with a *second head* that predicts the Q-values for the intrinsic reward; this branches off after the 'Post-LSTM linear layer' (with 256), and is the same type of dueling head, using the same scale normalisation method [Schaul et al., 2021]. In addition, the 5-step Q-learning is adapted to use a simple off-policy correction, namely trace-cutting on non-greedy actions (akin to Watkins Q($\lambda$) with $\lambda = 1$), separately for each learning head. The $\mathcal{X}_I$ policy is the greedy policy according to the Q-values of the second head. Note that because of these differences in set-up, and especially because the second head can function as an auxiliary learning target, it may be misleading to compare $\mathcal{X}_I$ and $\mathcal{X}_U$ results head-to-head: we recommend looking at how things change within one of these settings (across variants of intra-episodic exploration or the baselines), rather than between them.

## A.5 Homeostasis

The role of the homeostasis mechanism is to transform a sequence of scalar signals $x_t \in \mathbb{R}$ (for $1 \leq t \leq T$) into a sequence of binary switching decisions $y_t \in \{0, 1\}$ so that the average number of switches approximates a desired target rate $\rho$, that is , $\frac{1}{T} \sum_t y_t \approx \rho$, and high values of $x_t$ correspond to a higher probability of $y_t = 1$. Furthermore, the decision at any point $y_t$ can only be based on the past signals $x_{1:t}$. One way to achieve this is to exponentiate $x$ (to turn it into a positive number $x^+$) and then set an adaptive threshold to determine when to switch. Algorithm 1 describes how this is done in pseudo-code. The implementation defines a time-scale of interest $\tau := \min(t, 100/\rho)$, and uses it to track moving averages of three quantities, namely the mean and variance of $x$, as well as the mean of $x^+$.

---

**Algorithm 1** Homeostasis

---

**Require:** target rate $\rho$
1: initialize $\overline{x} \leftarrow 0, \overline{x^2} \leftarrow 1, \overline{x^+} \leftarrow 1$
2: **for** $t \in \{1, \dots, T\}$ **do**
3:     obtain next scalar signal return $x_t$
4:     set time-scale $\tau \leftarrow \min(t, \frac{100}{\rho})$
5:     update moving average $\overline{x} \leftarrow (1 - \frac{1}{\tau})\overline{x} + \frac{1}{\tau}x_t$
6:     update moving variance $\overline{x^2} \leftarrow (1 - \frac{1}{\tau})\overline{x^2} + \frac{1}{\tau}(x_t - \overline{x})^2$
7:     standardise and exponentiate $x^+ \leftarrow \exp\left(\frac{x_t - \overline{x}}{\sqrt{\overline{x^2}}}\right)$
8:     update transformed moving average $\overline{x^+} \leftarrow (1 - \frac{1}{\tau})\overline{x^+} + \frac{1}{\tau}x^+$
9:     sample $y_t \sim \text{Bernoulli}\left(\min\left(1, \rho\frac{x^+}{\overline{x^+}}\right)\right)$
10: **end for**

---

In our informed trigger experiments we use value promise as the particular choice of trigger signal $x_t = D_{\text{promise}}(t - k, t)$. As discussed in Section 3.1, when using a bandit, its choices for target rates are $\rho \in \{0.1, 0.01, 0.001, 0.0001\}$.

# B   Other variants

The results we report in the main paper are but a subset of the possible variants that could be tried in this rather large design space. In fact, we have done initial investigations on a few of these, which we report below.

## B.1   Additional explore modes

**Softer explore-exploit modes**   The all-or-nothing setting with a greedy exploit mode and a uniform random explore mode is clear and simple, but it is plausible that less extreme choices could work well too, such as an $\varepsilon$-greedy explore mode with $\varepsilon = 0.4$ and an $\varepsilon$-greedy exploit mode with $\varepsilon = 0.1$. We denote this pairing as $\mathcal{X}_S$. Preliminary results (see Figure 14) indicate that overall performance is mostly similar to $\mathcal{X}_U$, possibly less affected by the choice of granularity and triggers.

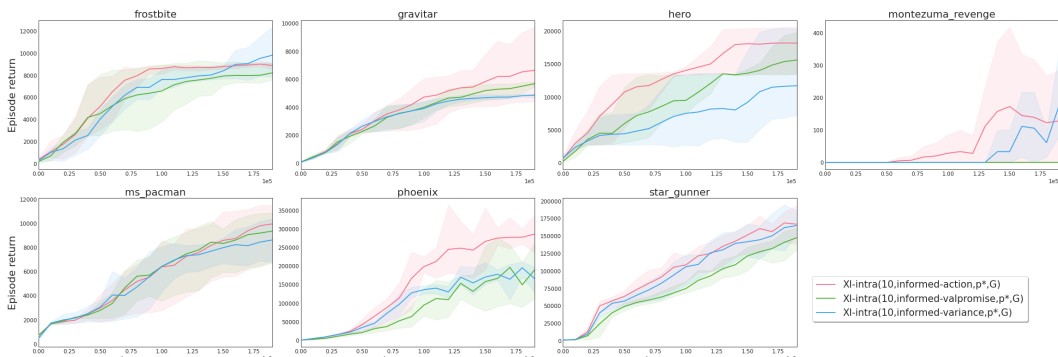

Figure 8: Preliminary results comparing different informed triggers: value-discrepancy, action-mismatch, and variance-based, when using $\mathcal{X}_I$ exploration mode.

**Different discounts** Another category of explore mode ($\mathcal{X}_\gamma$) is to pursue external reward but at a different time-scale (e.g., a much shorter discount like $\gamma = 0.97$). This results in less of a switch between explore and exploit modes, but rather in an alternation of long-term and short-term reward pursuits, producing a different kind of behavioural diversity. So far, we do not have conclusive results to report with this mode.

### B.2 Additional informed triggers

**Action-mismatch-based triggers** Another type of informed trigger is to derive an uncertainty estimate from the discrepancies across an ensemble. For example, we can train two heads that use an identical Q-learning update but are initialised differently. From that, we can measure multiple forms of discrepancy, a nice and robust one is to rank the actions according to each head and compute how large the overlap among the top-$k$ actions is.

**Variance-based triggers** Another type of informed trigger is to measure the variance of the Q-values themselves, taken across such an ensemble (of two heads) and use that as an alternative uncertainty-based trigger.

Figure 8 shows preliminary results on how performance compares across these two new informed triggers, in relation to the value-promise one from Section 2.4. Overall, the action-mismatch trigger seems to have an edge, at least in this setting, and we plan to investigate this further in the future. From other probing experiments, it appears that for other explore modes, different trigger signals are more suitable.

## C   Additional results

This section includes additional results. Wherever the main figures included a subset of games or variants (Figures 4, 5, 7) we show full results here (Figures 10, 11, 12, respectively), and the aggregated performances of Figure 3 are split out into individual games in Figure 9. Also, some of the learning curves from Figures 4 and 10 are shown in Figure 14. In addition, Figure 13 illustrates how the internal bandit probabilities evolve over time based on starting mode for the experiments shown in Figure 6.

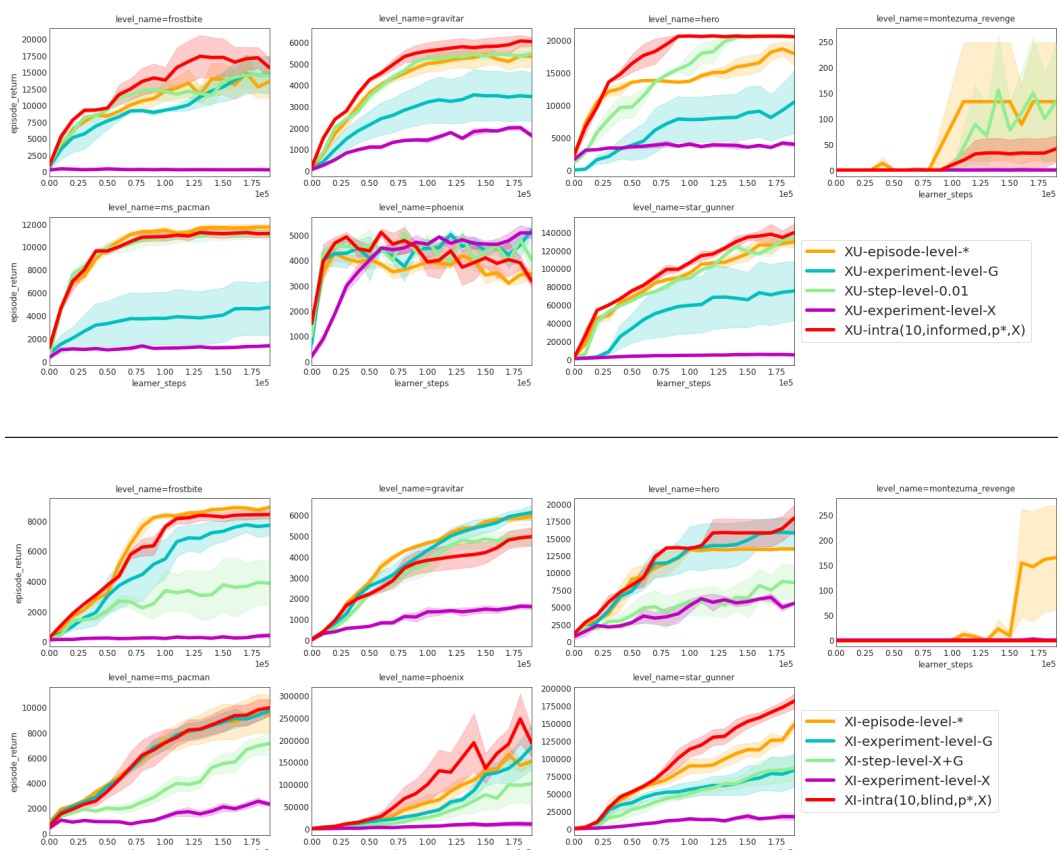

Figure 9: Extension of Figure 3, showing the characteristic space of exploration and how different explore-exploit proportions translate to performance, for $\mathcal{X}_U$ mode (top) and $\mathcal{X}_I$ mode (bottom).

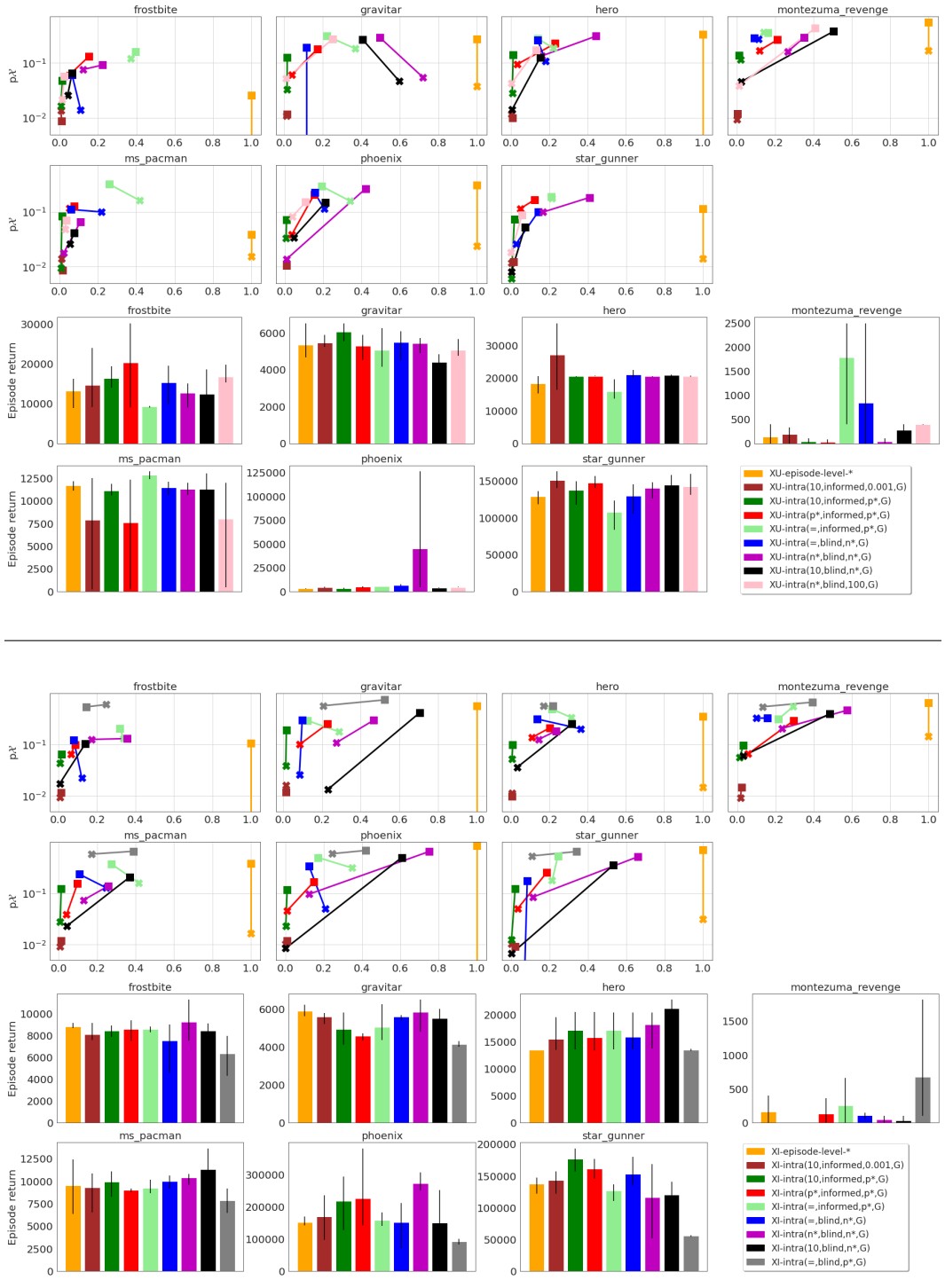

Figure 10: Extension of figure 4 for XU mode (top) and XI mode (bottom).

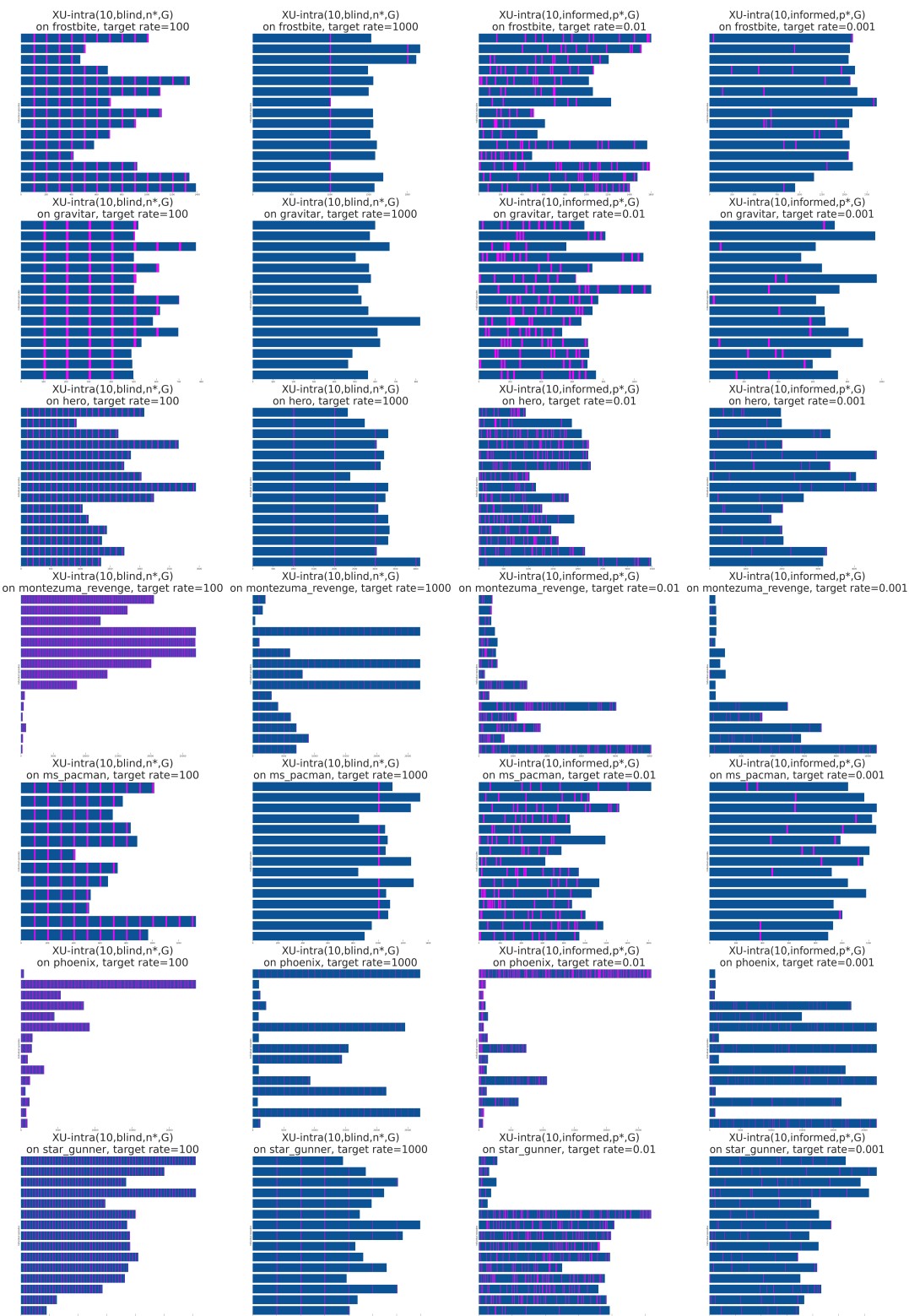

Figure 11: Extension of Figure 5 to the 7 Atari games we experimented with. **First two columns:** temporal structures for a blind, step-based trigger; the $15$ episodes we randomly selected correspond to $100$ and $1000$ fixed switching steps; the exploration period was fixed to $10$ steps. **Last two columns:** temporal structures obtained with an equivalent informed trigger and corresponding to target rates of $0.01$ and $0.001$, respectively.

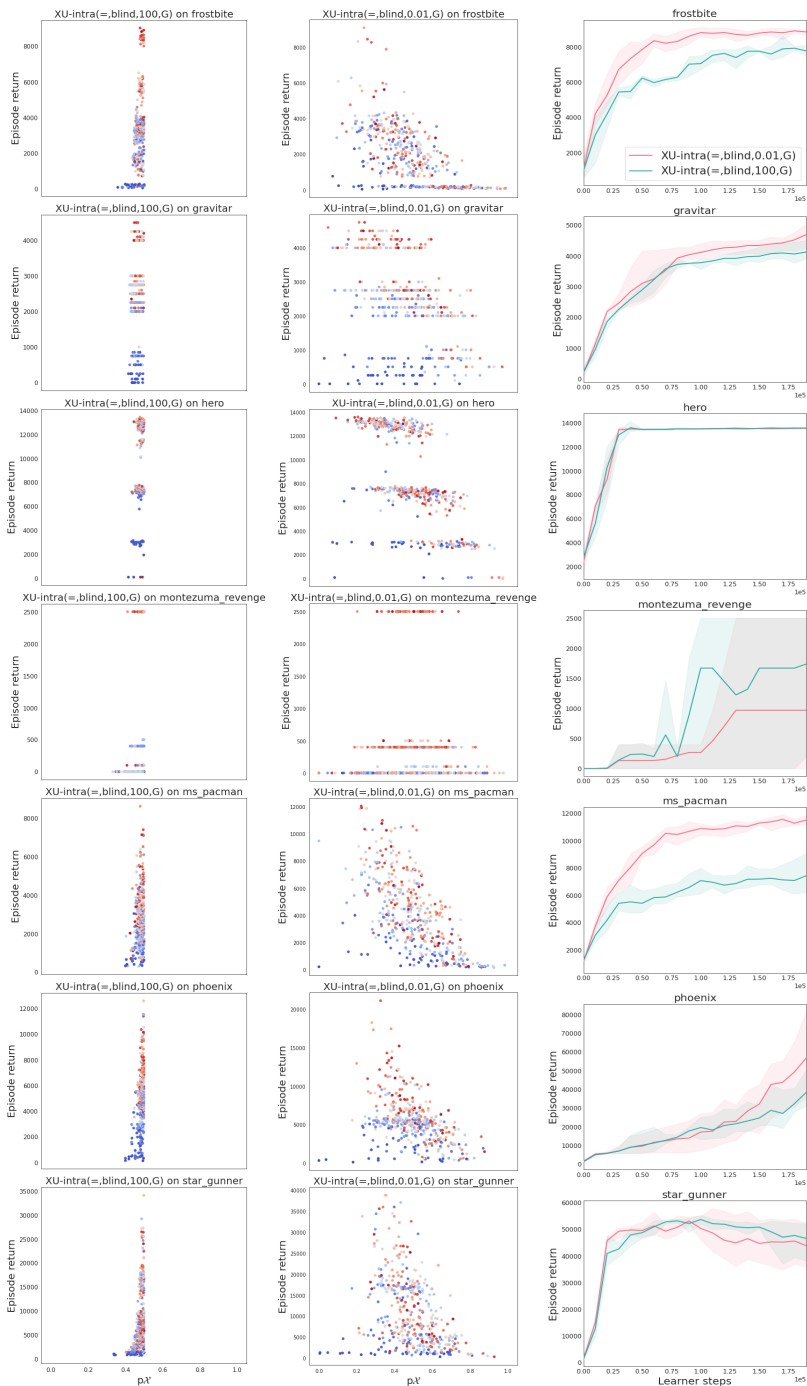

Figure 12: Extension of Figure 7, showing behavioural characteristics (exploration proportion $p_{\mathcal{X}}$) between two forms of blind switching, step-based (left) and probabilistic (center), with their corresponding performances (right).

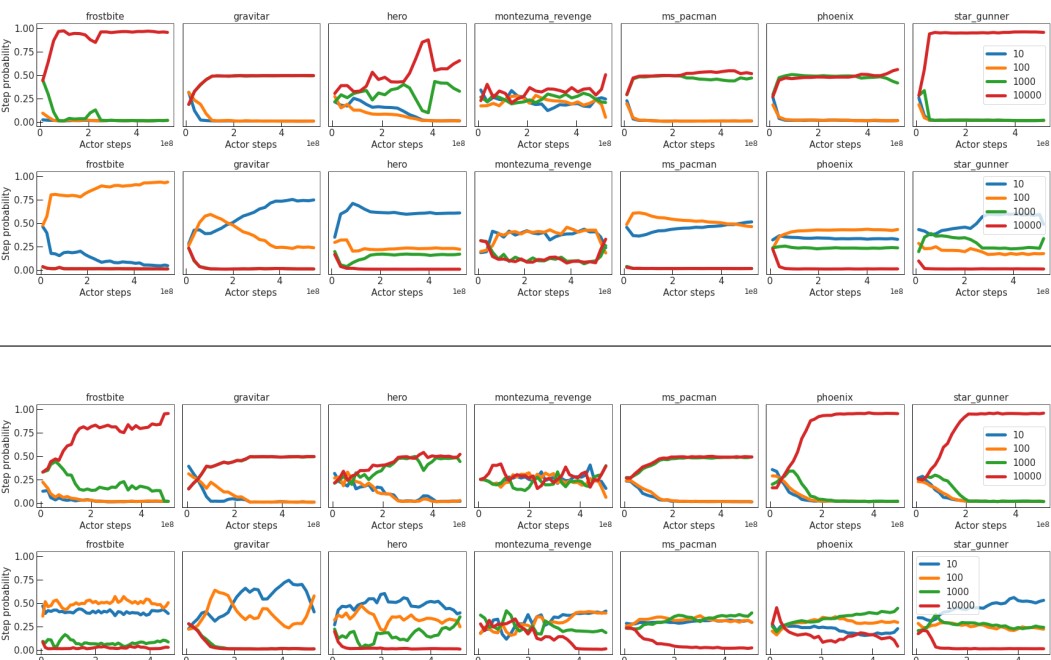

Figure 13: Extension of Figure 6, showing the performance differences between two blind intra-episode experiments, starting either in explore ($\mathcal{X}$, rows 2 and 4) or in exploit mode ($\mathcal{G}$, rows 1 and 3). We show the bandit arm probabilities for each of the step sizes $n_\mathcal{X}$ and how they change over the course of learning for $\mathcal{X}_U$ (top two rows) and for $\mathcal{X}_I$ modes (bottom two rows). **Findings**: for symmetric blind triggers, starting with exploitation results in slower rates of switching (high $n_\mathcal{X} = n_\mathcal{G}$ like red and green); in contrast, starting with exploration results in behaviours promoting higher switching rates (small $n_\mathcal{X} = n_\mathcal{G}$ like blue and orange). Note that these preferences are not matching perfectly across all games, and thus results are domain-dependent.

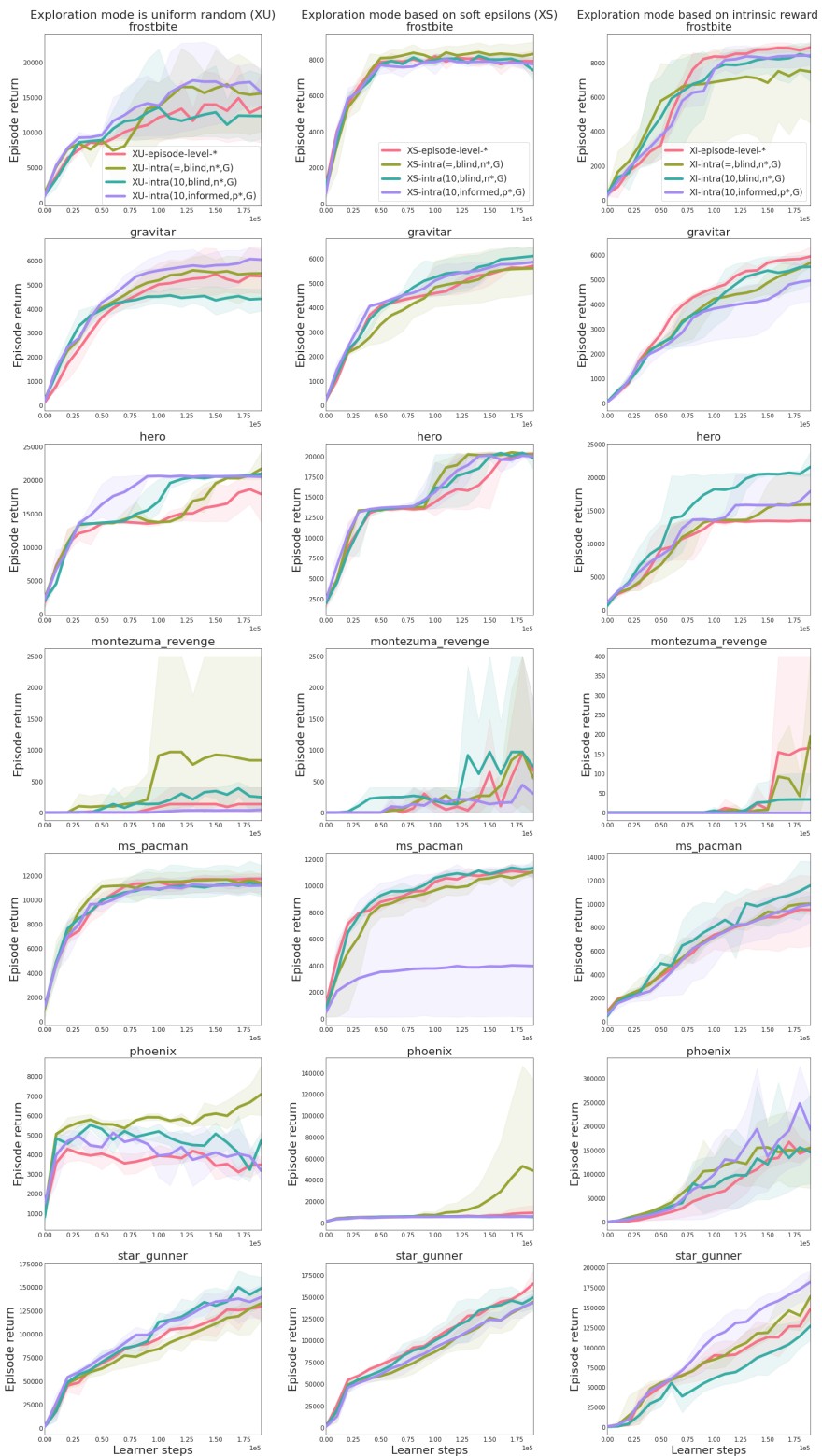

Figure 14: Comparing 3 different $\mathcal{X}$ modes on the same 4 experimental settings and across 7 Atari games: uniform exploration ($\mathcal{X}_U$, left), soft-epsilon-based exploration ($\mathcal{X}_S$, center), and intrinsic exploration ($\mathcal{X}_I$, right).