# OpenReview forum: "When should agents explore?"
_NeurIPS.cc/2021/Conference — NeurIPS 2021 Submitted_

### Official Review · Reviewer_h8V7 · 2021-07-16

**Rating:** 5
**Confidence:** 4

**Summary:**

The paper focuses on switching between exploration and exploitation modes in RL. Specifically, it examines the temporal granularity of exploration periods, decisions of when to switch between explore and exploit modes, and comparisons between different types of exploration. The work uses  a ‘value promise discrepancy’ for informed switching between modes. Experimental results are presented on a number of Atari games.

**Limitations And Societal Impact:**

I’d like to see more discussion of limitations. There are many variants of the method and parameters to tune, so the complexity of this approach might make use in other domains difficult.

Compute needs: there is no mention of the substantial compute required in the paper, and the section in Appendix A could be elaborated on. 2 Billion frames per run * 3 seeds * 5 methods * 2 settings = 60 billion frames, not including any experimentation or hyperparameter tuning. In addition to the environmental impact, this level of computation is not accessible to most researchers. Limitations of compute needs and lack of sample efficiency should be addressed.

The checklist says N/A for societal impacts; even if the authors don’t see the need to include a section in the paper, I’d like to see a justification of why this is the case.

**Main Review:**

Originality: The work focuses on the question of when to switch between different modes. This temporal aspect of exploration is frequently overlooked in prior work on exploration, which tends to focus more on the type of exploration. This temporal analysis is interesting and sheds more light on interpreting RL.

Quality: The work is technically sound, and the experimental results are detailed with many different visualizations. The Atari environments used are quite challenging, 5 of which are hard exploration games that prior work frequently struggles with or does not address.

Clarity: The work is generally well-written. I do think some parts of the paper could be reorganized for improved clarity. For example, sections 2.3 and 2.4 both cover switching and could be combined. Section 3.1 is very helpful for understanding the variants used. Some of the results could be better explained: figures 4 and 6 took quite a bit of time to understand, as these are visualization techniques unique to this paper. Other parts could have better analysis (see figure 3 and figure 5 notes below).

Significance: The paper brings temporal analysis to the forefront of RL, which could be interesting to many researchers. The impact could be greater with clearer take-aways. The overhead of using a metacontroller with many parameters to tune will make it challenging for researchers to incorporate the ideas. Clear take-aways (e.g. which fixed exploration strategy is best) that can be used without the entire meta-controller setup would allow many people to make simple tweaks to improve their RL work.

Minor comments:
Line 93 states that blind switching “does not take state or time into account” - but blind switching *only* depends on time.

Figure 3: I don’t understand why the experiment-level uniform exploration performance improves over time - shouldn’t this be a random policy, and therefore have flat reward curves? From the extension in the appendix (figure 9), it seems that Gravitar and Phoenix are the most increasing curves. Phoenix is especially surprising, where the pure exploration curve ends up with higher performance than all other methods. I’m also surprised that the experiment-level uniform exploration looks to be similar performance as experiment-level RND, and would like to see discussion of this. The large variance on these aggregate plots also makes it difficult to draw conclusions from (more than 3 seeds could help with this). I’d like to see analysis on why intra-episodic exploration performs worse than baselines on Montezuma’s revenge in both cases (appendix Figure 9).

Figure 5: The illustration of temporal structure is really interesting. I’d like to see more analysis on this, for example what scenarios happen during the dense periods of exploration.

Figure 6: I’d be curious to see the analogous results for non-blind runs. Also, could the meta-controller decide whether to start in explore or exploit mode? Given that which is better is heavily game-dependent, this seems like something well-suited to a meta-controller.

Figure 7: A colorbar legend would be helpful here, especially for readers who are colorblind or reading a black and white printed copy.

Related work: This section is quite short. I’d like to see more situating of this paper relative to prior work on non-temporal exploration. A section on bonus-based exploration would be helpful.

Additional methods: I’d like to see a comparison to ε decay (instead of fixed ε in the ε-greedy baseline), since this is commonly used in RL work. I’d also like to see the intrinsic results where exploit mode does not include the auxiliary task (modes are entirely separate), as this might be a better comparison to the uniform exploration cases.

**Time Spent Reviewing:**

4

---

> ### Author Response · Authors · 2021-08-10
> **Response to reviewer #4**
>
> We thank Reviewer 4 for their in-depth review, based on which we have improved the clarity of the figures and explanations in our paper. We would like to respond to the following remarks:
>
> * “there are many variants of the method and parameters to tune”: to clarify, our **meta-controller has zero hyper-parameters** of its own, so its addition directly compensates the added tuning complexity of the additional degrees of freedom we introduce (trigger rates and exploration ratios).
> * “the checklist says N/A for societal impacts; even if the authors don’t see the need to include a section in the paper, I’d like to see a justification of why this is the case”: we view our work as being fundamental research, not ready for immediate use, so clear societal impacts cannot be pinned down at this point. We will clarify this in the updated version.
> * “the impact could be greater with clearer take-aways (e.g. which fixed exploration strategy is best)”: we thank the reviewer for the opportunity to clarify that there is no fixed exploration strategy that is the best. Each domain has a different preferred sub-episodic exploration scheme. Our message is that a fine-grained view of behaviours and switching between them translates to rich, diverse, and beneficial behaviours. On top of this, the meta-controller is a strength, since it allows the agent to adapt the “when” answer throughout training. Thus, we view the meta-controller as a convenient, easy to integrate component, relieving us from tuning parameters or recurring to heuristics. For these reasons, we think our method is suited for practitioners too.

---

### Official Review · Reviewer_SAR7 · 2021-07-16

**Rating:** 5
**Confidence:** 3

**Summary:**

This paper proposes several key parameters for studying when to switch between exploration and exploitation modes in hard-exploration RL problems. These parameters include frequency and timescale for exploration mode, how the mode is triggered, and how exploration is performed. The authors show that in certain circumstances, correctly setting these parameters can lead to significant gains in task performance, and discuss methods for setting these parameters without significant hyperparameter tuning.

**Limitations And Societal Impact:**

The authors address the limitations of this work, but it would be helpful to include a discussion about its social impacts.

**Main Review:**

Overall. Learning when to explore is an important problem for meta-learning or continual learning. The parameters explored in the paper are presented well, although it would be helpful to show more random seeds. The paper discusses many interesting properties of intra-episode exploration, but ultimately the effect of these properties on task performance appears inconclusive.

Pros:
1) The paper was well organized and the writing was clear.
2) Experimental parameters are clearly defined, and ablations are thorough.
3) There are some surprising results, such as the importance of starting mode and of step based vs probabilistic switching.

Cons:
1) While learning when to explore is an interesting problem, in the setting of the paper (Hard-exploration Atari games with up to 1B frames' data) there are many other methods that are relevant and comparable [1, 2, 3]. In particular, it would be interesting to see comparisons to a method that simultaneously uses intrinsic/exploration and extrinsic rewards.
2) I would be interested in seeing more seeds per experiment.
3) The paper focuses on the potential benefits on intra-episode mode-switching, but it doesn't appear that this method significantly outperforms episode-based switching overall.

Questions:
1) In Figure 3, why are the XI-experiment-level-G curves different between the left two and right two plots? Does exploration mode affect the greedy policies?
2) Do you have an intuition about why the blind trigger may outperform the intrinsic one, as in Phoenix, or when uniform exploration may outperform intrinsic motivation?
3) Could you provide some further discussion about how option-based exploration relates to this work?

Details:
- (Section 3.1 Line 169) - end -> and
- (Figure 5 and 6) - Are different seeds used for these experiments?

[1] Ecoffet, A., Huizinga, J., Lehman, J. et al. First return, then explore. Nature 590, 580–586 (2021). https://doi.org/10.1038/s41586-020-03157-9

[2] Y. Burda, H. Edwards, A. Storkey, O. Klimov. Exploration by Random Network Distillation.

[3] J. Martin, S Narayanan S., T Everitt, M Hutter. Count-Based Exploration in Feature Space for Reinforcement Learning


**Time Spent Reviewing:**

3

---

> ### Author Response · Authors · 2021-08-10
> **Response to reviewer #3**
>
> We thank Reviewer 3 for their thorough reading of our paper and great questions, for which our responses are:
> * “In Figure 3, why are the XI-experiment-level-G curves different between the left two and right two plots? Does exploration mode affect the greedy policies?”: yes, they are different, because the XI and XU settings differ: specifically, in XI, we train 2 heads instead of 1. We mentioned this in the second paragraph of section 3.2.
> * “Do you have an intuition about why the blind trigger may outperform the intrinsic one, as in Phoenix, or when uniform exploration may outperform intrinsic motivation?”: This is a great observation – the blind trigger that performs the best for Phoenix is a symmetric trigger, which means that it is guaranteed that there is roughly a 50-50 exploration-exploitation chance in the beginning of training (first 10M frames) when the bandit prefers num_step=10, so there is roughly equal amounts of exploration and exploitation. So, the blind bandits are more aggressive in terms of the imposed amount of exploration, which, in the case of Phoenix, also accounts for higher performance.
> * “(Figure 5 and 6) - Are different seeds used for these experiments?” Yes, in all our figures (including Figures 5 and 6) we extracted data and performed our analysis based on all 3 seeds we ran. When we show error bars, they are between the minimum and maximum values across our 3 seeds. In Figure 5, we did not control the seed when selecting the 15 random episodes for which we analysed the temporal structure. In Figure 6, the aggregation is performed across the 3 seeds.
> * “it would be interesting to see comparisons to a method that simultaneously uses intrinsic/exploration and extrinsic rewards”: we would like to clarify that, in Figure 3, the XI-step-level X+G experiment (light green curve) uses precisely that setting and is also very similar to Burda et al. in reviewer's reference [2].
> * “it doesn't appear that this method significantly outperforms episode-based switching overall”: we view intra-episodic switching as a desirable and useful component of an agent's learning process. Our intra-episodic scheme endows an agent with a new feature: to be flexible and adaptive, coarse or fine-grained since it can (learn how to) choose (on the fly) the appropriate exploration granularity adjusted to the training domain and moment in time.
> * We provide some initial experimentation on a well-known benchmark (hard exploration Atari games), but this is nevertheless the beginning of our journey. While we respect it, we disagree with the reviewer's claim that "ultimately the effect of these properties on task performance appears inconclusive". We might not have found a single best intra-episodic setting that surpasses the baselines across all Atari games tried (since that was not the focus of our current work), but this does not deny the usefulness of our method for performance -- there is at least one intra-episodic variant for each game which results in clear performance gains (just that it is not the same intra-episodic variant across games). We are also confident that the performance gap will become even more significant for (1) more complex explore-exploit modes; (2) more challenging and naturalistic domains.

---

### Official Review · Reviewer_v6F4 · 2021-07-19

**Rating:** 4
**Confidence:** 5

**Summary:**

This paper studies the question of whether there is value in switching between exploration and exploitation behavior in RL. The authors investigate different signals for switching between learning a policy which maximizes an intrinsic reward (RND) and exploiting an extrinsic reward. The authors evaluate their approach on a number of Atari games, and show that an intra-episodic exploration scheme that switches based upon a "value discrepancy" is comparable/better to "episode" level exploration scheme on some evaluations. The authors additionally contribute some analysis showing the qualitative switching behavior and its effect on downstream performance.

**Limitations And Societal Impact:**

The authors also raise the concern of the problem of "off-policy-ness" in their limitation section. While I feel they understate the difficulties of addressing this issue, I feel that their self-evaluation of their limitations is reasonable.

**Main Review:**

Originality

This paper investigates a setting that is not commonly studied in the RL literature (namely one where exploration and exploitation are explicitly decoupled). While I did not find the particular "value promise" trigger to be well motivated in the writing, to my knowledge such a scheme is novel to the best of my knowledge.

Quality

Overall, I feel that this paper provides some good exploratory experiments which suggests that there is some merit to this idea. However, I think there are some high level conceptual/experimental issues which I feel need to be resolved before publication.

There are some clear conceptual issues with how this paper explicitly decouples the exploration and exploitation schemes in RL from an optimization perspective. The main one being that when done with a novelty seeking bonus such as RND, the data produced by exploration to be very off-policy from the current exploitation policy. This causes me to be skeptical of the overall stability/scalability of this type of approach, even with the addition of off-policy correction terms. From a practical perspective, this paper also introduces a layer of complexity in determining the switching procedure between this behavior mode. The authors attempt to address this with a bandit based  controller, but given somewhat incremental improvements in performance I do not feel this additional complexity is justified.

From an experimental perspective, another concern I have is that this paper does not compare to other approaches such as Bagot et al., 2020 which also proposes a similar high level scheme. While this paper is cited in the related work (among other papers), the authors neither provide a comparison nor give a reason why a comparison is not appropriate.

Clarity

Overall, I found the exposition in this paper to be somewhat opaque. I was not able to figure out important experimental details from reading the main paper (e.g., the underlying RL algorithm). I also found the captions and plots a little difficult to parse (e.g., Fig 3, all of the plots share the same title). Finally I would appreciate a dedicated related work section beyond the single paragraph provided. I feel it is the author's duty to situate their contribution in the literature clearly, but I felt in the current writing this is distributed throughout the paper in an unnecessary way.

Significance

I feel that this paper starts with an interesting premise, but falls short in delivering clear evidence that 1) the exploration scheme proposed is clearly superior to conventional exploration schemes, 2) such an approach would not result in the need for significant off-policy correction, 3) the method proposed here is better than other similar schemes proposed previously. Due to this, I do not believe this paper in its current form is ready for publication without significant improvements.

**Time Spent Reviewing:**

3

---

> ### Author Response · Authors · 2021-08-10
> **Response to reviewer #2**
>
> We thank Reviewer 2 for their valuable comments and thorough feedback. We address their concerns in order:
> * “the approach would need significant off-policy correction”: this is an excellent point, thank you for raising it! As discussed in the paper, we are also concerned about this as a limitation, but prodded by your comments, we dug a bit deeper into the question during the rebuttal period. Specifically, we ran a control experiment for intra-episodic (XI) **without any off-policy correction**, and found that performance is the same (or even slightly better). So, surprisingly, **off-policy correction is not essential** in our setting! We will provide and discuss these results in the updated version of our paper. While this new evidence does not void the off-policy concern in general (we agree with the reviewer this is a serious issue in the medium-term), it shows that it is not critical in the current setting yet.
> * “The exploration scheme proposed is not clearly superior [...] the additional complexity [...] is not justified”: we believe that our methodology, idea, and qualitative investigation were interesting in and of themselves. We did not intend to “solve” intra-episodic switching and make it ready for real-world applications, but merely to lay out the initial steps as we see in this a promising, novel direction for exploration research. In terms of extra complexity, while we introduce a number of additional design choices, we simultaneously provide a (hyper-parameter-free) meta-controller that prevents them from becoming a tuning burden.
> * “dedicated related work section beyond the single paragraph”: we cited many and varied research studies across the paper, summing up to **4 pages of references with 74 citations in total**, but we will happily include any additional citations the reviewers suggest adding.
> * “does not compare to other approaches such as Bagot et al., 2020”: this paper is indeed related to ours, in particular, it shares the idea of switching in and out of exploratory phases. A main difference is how exploratory phases are initiated: this is done by augmenting the actions space with a special “explore-from-here” action that the RL agent can learn to use (instead of our triggers). Also, it uses a different benchmark domain (tabular grid-world vs Atari) and places less emphasis on the “when” aspects of exploration. We will expand on this comparison in our final version.

---

> > ### Comment · Reviewer_v6F4 · 2021-08-26
> > **Thanks for your response**
> >
> > I've read the authors response. I have kept my current score and provide my rationale to follow. First, I am still quite skeptical of the difficulty of applying such an approach without any off-policy correction. I see no intrinsic reason why the exploration behaviour should not be extremely divergent from the current policy, which a priori I would expect to lead to poor performance or even extreme instability. Is there a way to control for this? Could the current level of performance be obtained without tackling this separate research problem? Second, while (Bagot et al., 2020) may have considered a different domain, I do not believe that it is a sufficient rationale to at least provide a comparison. I would recommend that the authors at least include a best effort re-implementation for any resubmissions. Finally, the number of citations and pages of references was not really my gripe with the current draft of the paper. It is the author's duty in my view to explain how their own work is situated in the literature/current state of knowledge at the time of submission. In my view, only including a large list of papers without discussion significantly detracts from the overall clarity of the work's contribution.

---

> > > ### Author Response · Authors · 2021-08-27
> > > **empirical off-policy resilience, stylistic choices**
> > >
> > > Thank you for reiterating your concerns, and allowing us another attempt to address them.
> > >
> > > 1) **Off-policy**: we did actually share your skepticism on the difficulty of off-policy learning with such radical switches in behaviour, when we initially embarked on this project. However, the intuition did not come to pass: in our empirical results the (far more) off-policy nature of experience **does not harm** agent performance, nor is there instability. Additionally, as we stated in our previous response: we ran an additional control experiment **removing** the off-policy correction from the XU setting (reverting to untruncated 5-step Q-learning), and found that performance is **the same or better**. This may again be counter-intuitive, but we have to accept the results as they are. One possible explanation is that multi-step Q-learning as a core algorithm is actually surprisingly resilient to off-policy data. Of course, it is possible that improving off-policy corrections could further improve performance of our method, but we do not see why that should count against our paper in its current form?
> > >
> > > 2) **Related work**: we take our duty seriously to situate our work within the literature. It appears that the reviewer takes issue with our **stylistic** choice of interleaving this with the main body of the paper (where related ideas first appear), instead of grouping the discussion of related work in a dedicated section? In fact, if we were to rearrange the sentences that discuss and contextualise all 74 cited papers, this would fill over a page. The concrete thing we could suggest to help alleviate this concern/confusion is to relabel the currently contentious paragraph as "**Additional** related work"?

---

### Official Review · Reviewer_XEyz · 2021-07-20

**Rating:** 4
**Confidence:** 4

**Summary:**

The paper argues that for effective exploration, switching behavior modes is important. Specifically, the method proposed switches behavior from exploit to explore for chunks within an episode.

**Limitations And Societal Impact:**

Limitations discussed above

**Main Review:**

**Significance** : While it is true that exploration is an important problem, and also that monolithic exploration is not a great approach, it is unclear if we really need sub-episodic switching. This is since switching exploratory behavior at the episodic level already works quite well. In fact on the Atari environments evaluated, episodic switching actually performs much better that the proposed approach on Montezuma's revenge (from training curves in the appendix, as well as the bar plot in figure 4), and is not significantly worse on the other Atari games. As such the main emphasis of the paper seem to be on the important of switching behavior within an episode, and I don't think the result show it to be that much more important that episodic switching.

**Originality** : The idea for the exploration trigger to be based on TD error is not very novel and has been studied in prior work [1]. Furthermore this is similar in spirit to prior work where exploration is done in regions of epistemic uncertainty. The paper does discuss related work in related to exploration with 'non-trivial temporal structure' as well as some work related to options, but this is quite brief, and doesn't include a comprehensive list of works for exploration in RL.

**Clarity, quality**: The ideas in the paper are well presented and easy to follow. While the authors test on Atari, they don't include any evaluation on continuous control environments, such as simulated robot environments, and so it's unclear if the insights are applicable in that domain. The main issue remains that it's unclear if sub-episodic exploration helps performance, given the results in the paper.

[1] : Reward Prediction Error as an Exploration Objective in Deep RL (IJCAI 20)

**Time Spent Reviewing:**

4

---

> ### Author Response · Authors · 2021-08-10
> **Response to reviewer #1**
>
> We thank the reviewer for their thoughtful comments, and for raising a number of interesting points. We believe that we can address all of them:
> * “it's unclear if sub-episodic exploration helps performance”: it is true that episodic switching is a strong baseline, but Figure 3 shows that sub-episodic exploration **helps on average**. Zooming in on individual games, we find among others that two intra-episodic variants, namely XI-intra(=,blind) and XI-intra(=, informed), **outperform episodic switching in Montezuma** too (see bar plots of Figure 10 in Appendix C).
> * “trigger to be based on TD error is not very novel”: while exploration methods driven in some form by TD errors (or other forms of prediction error) are indeed common, **no prior work has used it to trigger a mode switch**. You could say that we build on the insights of the rich prior work, but use the ideas in a different and novel way; we will clarify these links in the revised paper.
> * “include a comprehensive list of works for exploration in RL”: we consider that our **four pages of references** demonstrate a quite comprehensive scholarly approach, with citations integrated throughout the text. But we are more than happy to expand on this, and would ask the reviewer which additional references they would like to see (in addition to [1] of course) so that we don’t miss anything important.
> * “include an evaluation on continuous control”: such environments are common benchmarks for deep RL, but generally **don’t exhibit a complex episodic structure**, for which judicious switching would be required: their main exploration difficulty is often related to the action space and control. Our method is not designed for such settings, and we speculate that it would therefore not help much. Thank you for raising this point though, we will add a discussion of this and more generally what domain our family of methods is appropriate for.

---

### Decision · Program_Chairs · 2021-09-27

**Decision:**

Reject

**Comment:**

The paper focuses on an exploration question the authors consider under-explored: "when to explore". Whereas some other exploration methods are either 'monolithic' (term used in the paper for strategies where the action selection at each timestep has both exploration and exploitation characteristics), or decides to use a purely explorative or exploitative behavior for either a single time step or a whole episodes, the authors propose a strategy for switching within an episode to stretches of an 'exploit' mode and an 'explore' mode.

The review process did not have a lot of interaction between reviewers and authors, such that I have taken a closer look at the paper itself and at the replies of the authors to determine which of the points raised by the reviewers have been sufficiently addressed.

On originality, the reviewers mostly agreed that the question of 'when' to explore was a new perspective, though one reviewer didn't find the approach itself wholly original.

On technical quality, there were several comments raised by the reviewers. One is that the experimental environments are narrowly chosen: all experiments are on Atari games, which have strong commonality on certain characteristics (e.g., being (close to) deterministic). Another is that potential benchmarks are missing: although some of the benchmarks are used a lot in practice, a strategy like epsilon-greedy is not a particularly strong baseline. Instead, a 'monolithic' exploration method could have been used (e.g. a combination of intrinsic and extrinsic rewards as suggested by a reviewer, or perhaps posterior sampling).
Moreover, I found the motivation for both mode-switching based exploration and the particular implementation (e.g. the proposal for 'value promise trigger') on the weak side. Many 'monolithic' methods (e.g. posterior sampling) would implicitly also switch between more 'exploration focused' or 'exploitation focused' behavior, why is switching between 'pure' exploration or exploitation important? The value promise trigger would I think trigger with either a non-converged value function (appropriately) or when the environment is intrinsically very random (where more exploration might not be sensible). I think the Atari environments are not a good test-bed as they are (almost) deterministic. Furthermore, at the moment the value trigger is high, the problematic state might be k steps ago.

On significance, most reviewers considered the comparative results between episode-level switching and the proposed strategy unconvincing. Here, the authors argued that (1) on all games, some intra-episode switching does perform best (2) there is some improvement overall on all games and (3) even when performance is not significantly better, intra-episode switching is still interesting. There is some merit to all of these argument, though the improvement is small overall and the variance is high (also because a relative low number of seeds is used). I find the third argument most interesting: in continuing tasks, one doesn't have the 'luxury' to re-set and thus intra-episode switching would be a good alternative to the now-impossible episode-level switching.

The writing was mostly considered clear by the reviewers though there were some concrete comments for improvement. Some confusion about the meaning of Fig 3 was not completely resolved by the authors reply.

All in all, I think that although the paper does present a new viewpoint, it is currently not clear enough how significant this contribution is in the state of the field, as experimental evidence is somewhat inconclusive and the motivation of design decisions is unclear.

Minor issue:
I think there are a number of issues with the use of gamma around line 120. States farther in the future from t-k are less this counted than closeby states, and the value at t isn't discounted at all.